# Crystal Growth, Luminescence and Scintillation Characterizations of $Cs_2KLaCl_6$:Ce and $Cs_2KCeCl_6$

**Haoyu Wang, Jianhui Xiong, Man Li, Jufeng Geng, Shangke Pan and Jianguo Pan ***

State Key Base of Functional Material & Its Preparation Science, Key Laboratory of Photoelectric Detection Materials and Devices of Zhejiang Province, Institute of Materials Science & Chemical Engineering, Ningbo University, Ningbo 315211, China; all18758625292@gmail.com (H.W.); yuanchennbu@163.com (J.X.); 13239103960@163.com (M.L.); 18829252234@163.com (J.G.); panshangke@nbu.edu.cn (S.P.)
* Correspondence: panjianguo@nbu.edu.cn

**Abstract:** Elpasolite halides scintillation crystals have been proven to be very important materials for X-ray and γ-ray detector applications. The crystals of $Cs_2KLaCl_6$:4% Ce (CKLC) and $Cs_2KCeCl_6$ (CKCC) belong to novel scintillation of the Chloro-elpasolite crystal family. In this paper, the crystal growth of CKLC and CKCC crystals using the vertical Bridgman techniques were reported. The PXRD patterns showing both crystals have a cubic crystal structure. Both crystals have similar photoluminescence excitation and emission spectra, the fluorescence decay time of CKLC and CKCC crystals were about 49.7 and 33.8 ns. The energy resolution under the excitation of 662 keV γ-rays from a $^{137}$Cs source were found to be 6.6% and 5.2% (FWHM), respectively. The scintillation decay times of CKLC crystal were $\tau_1 = 127$ ns (33%) and $\tau_2 = 1617$ ns (67%), while that of CKCC crystal were $\tau_1 = 2.86$ ns (5%) and $\tau_2 = 81$ ns (95%).

**Keywords:** elpasolite; Bridgman technique; photoluminescence; scintillator materials

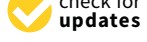

## 1. Introduction

Scintillation crystal is widely used in security inspection, medical treatment, petroleum exploration, non-destructive inspection and other fields as component materials of radiation detection equipment [1–9]. In the last two decades, a large number of new halide scintillation crystals with excellent scintillation properties have been found, among which the elpasolite halides scintillation crystals with structural composition of $A_2BMX_6$ (with A = Rb, Cs; B = Li, Na, K; M = La–Lu; X = F, Cl, Br) have good light output and excellent energy resolution, which has attracted people's attention [10–13]. Compared to other scintillators, elpasolite halide scintillators generally have higher energy resolution. On the other hand, elpasolite crystals have cubic or quasi cubic isotropic structure, and were seldom affected by thermal mechanical stress and cracks during crystal growth, therefore, the high-quality single crystals are easy to grown [14], such as $Cs_2LiYCl_6$:Ce, $Cs_2LiLaCl_6$:Ce, $Cs_2NaLaCl_6$:Ce, $Cs_2NaLaBr_6$:Ce [15–17]. Through isomorphic replacement and solid solution methods, the elpasolite crystal family will produce many new crystals. The current research mainly focuses on the discovery of new pasolite family crystals with high energy resolution and fast decay time [18].

In this report, the crystal growth, luminescence and scintillation characterizations of newly CKLC and CKCC single crystal are reported. Photoluminescence, X-ray induced luminescence, pulse height and scintillation decay time spectra are presented. Compared with other crystals in the elpasolite crystal family, the CKLC and CKCC crystal have smaller energy resolution, faster decay time. The study shows that both materials are a promising scintillator in the field of radiation detection especially CKCC crystal.

## 2. Experimental Section

### 2.1. Synthesis and Crystal Growth

Polycrystalline materials of CKLC (with 4 mole% of $Ce^{3+}$) and CKCC for crystal growth were prepared by melting the raw materials in quartz ampoules. The stoichiometric amounts of anhydrous CsCl (4 N), KCl (4 N), $LaCl_3$ (4 N) and $CeCl_3$ (4 N) from Beijing Grinm Advanced Materials Co. Ltd. were weighed in an $N_2$ (99.99%) purged glove box which has moisture and oxygen levels no more than 1 ppm. The weighed powder was put into a dry quartz ampoule having an inner diameter of 12 mm. The quartz ampoule was previously washed with HF and deionized water, and then baked in a vacuum at ~180 °C. To drive out the residual moisture, the mixed raw materials were baked in a vacuum at about 200 °C for several hours and sealed under dynamic vacuum of $\sim 10^{-1}$ Pa using an oxy-acetylene torch.

The single crystal of CKLC and CKCC was grown through the vertical Bridgman technique. First, the powder in the sealed quartz ampoule was melted in a muffle furnace and kept at this temperature for 24 h, to ensure that the components are mixed evenly and reacted fully. Then, the quartz ampoule was transferred to the vertical Bridgman furnace with the temperature gradient of about 20 °C/cm. During the crystal growth process, the temperature of the furnace was set to 20 °C above the melting point and monitored by Pt-Pt/Rh thermocouples, the pull rate was fixed at 8 mm/d, after growth, the ampoules were cooled down to room temperature with a cooling rate of 30 °C/h. Cutting, grinding and polishing of the crystals was performed in a dry room, the samples of CKLC and CKCC crystals were saved in mineral oil in a glove box.

### 2.2. Characterization

The Bruker D8 Focus X-ray powder diffraction technique (Cu Kα1 radiation) with an X-ray source operated at 40 kV voltage and 10 mA current was used to perform the test of the sample. The scanning ranger of two-theta angle was from 10° to 80° with a step size of 0.02°. The lattice constants were calculated by MDI Jade 6.0 software. TG and DTA curves were obtained utilizing the NETZSCH STA2500-0255-N instrument to detect phase change and melting point of the sample. Under the protection of nitrogen atmosphere, the powder samples in an alumina crucible were heated from 20 to 650 °C at the heating rate of 10.0 K/min.

The steady-state photoluminescence (PL) spectra of studied samples was measured by a HITACHI F-4600 spectrophotometer (Hitachi, Japan) using a xenon lamp as the excitation source at room temperature. A self-assembled X-ray spectrometer with a tungsten target was used to obtain the X-ray induced luminescence spectrum of CKLC and CKCC crystals in the wavelength range of 300–500 nm. Among them, the working voltage and current of the X-ray tube are 50 kV and 0.5 mA, respectively. The photoluminescence decay curve was obtained by HORIBA FL3-111 fluorescence spectrometer with a pulse width of 1 ns. Pulse height spectrum of samples of CKLC and CKCC crystals were measured under the excitation of 662 kV γ-ray from the [137]Cs source. The photons released by the crystal were received by a photomultiplier tube (PMT, R2059, Hamamatsu), amplified and converted into electrical signals, and finally the electrical signals to obtain pulse height spectrum processes. The shaping time was set to 3 μs, and the high voltage of the PMT was set to –950 V. The test of scintillation decay time under the excitation of gamma ray was completed by a device independently built by the Shanghai Institute of Silicic Acid. The excitation source was [137]Cs, and the scintillation decay time was read using an oscilloscope (Tektronix DPO 5104). Before the test, the crystal was wrapped with a Teflon film, leaving only one side as the light collecting surface.

## 3. Result and Discussion

### 3.1. Crystal Growth

Single crystals of CKLC and CKCC were obtained successfully through the vertical Bridgman technique. CKLC crystals are shown in Figure 1a. The crystals of the shoulder

area had excellent transparency and were free of cracks, at the same time, the upper parts of the crystals in the ampoules were polycrystal; this situation may be caused by the upward movement of impurities during single crystal growth. Several surface defects were observed in CKLC crystals as a result of adhesion to the quartz, but these defects are easily polished away. The polished samples of CKLC with the thickness of 10 mm are shown in Figure 1b. Figure 1c show the grown CKCC crystal in quartz ampoule. The bottom area of the crystal has a translucent appearance, the shoulder area of the CKCC crystal has a clear appearance and is free of surface defects. A crack and inclusion free boule with the thickness of 10 mm is shown in Figure 1d.

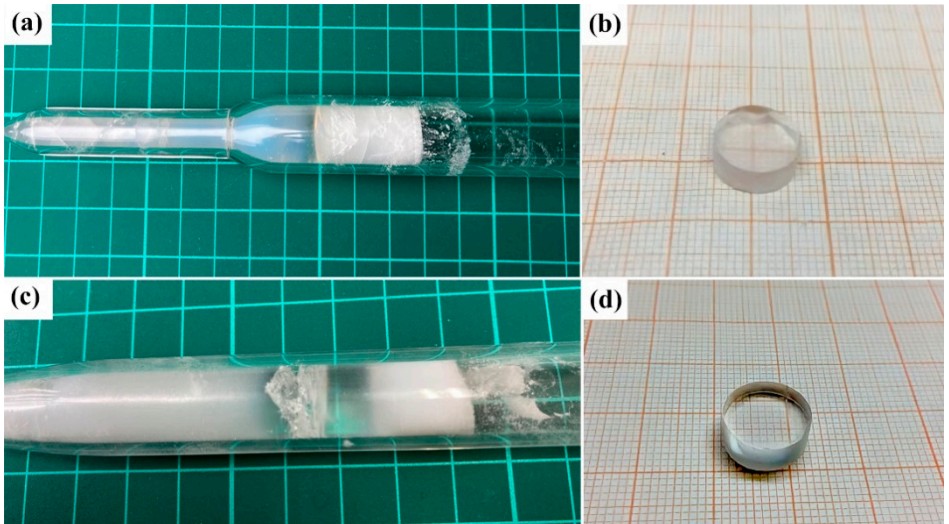

**Figure 1.** Photographs of (**a**) CKLC and (**c**) CKCC crystal in an ampoule, photographs of (**b**) CKLC and (**d**) CKCC polished crystal samples.

According to our long-term experience in crystal growth, ideal transparent crystals usually appear in the shoulder area of the ampoule by the Bridgman method. The transparency of the bottom area of the crystal is slightly lower than that of the shoulder area, and cracks may appear. The crystal in the top area is polycrystalline or opaque. This phenomenon can be seen from Figure 1. The deviation of the temperature setting and the choice of the shape of the ampoule may cause this to happen. In future research, we will use other shapes of ampoules to grow CKLC and CKCC, and find the most suitable ampoules for CKLC and CKCC growth.

### 3.2. Crystal Analysis

In order to verify this judgment, we performed powder XRD tests on samples of these three areas of two ampoules. The powder X-ray diffraction patterns of CKLC and CKCC crystals are represented in Figure 2. In order to avoid the interference of moisture during the PXRD measurement, the powder samples were sealed in a glass box. The XRD patterns of different parts of the CKLC and CKCC crystal are identical, and there are no extra peaks, indicating that the composition of the upper, middle and lower parts of the crystal is consistent. The PXRD data confirm that both crystals are cubic structures with Fm-3m space group. The cell parameters of these crystal are determined to be: CKLC: $a$ = 11.127 Å, V = 1377.8 Å$^3$, d = 3.17 g/cm$^3$, $Z_{eff}$ = 49 and CKCC: $a$ = 11.084 Å, V = 1361.7 Å$^3$, d = 3.20 g/cm$^3$, $Z_{eff}$ = 50. The higher density or effective Z number will help increase the stopping power and count rates under the excitation of $\gamma$ rays. The peaks of CKLC and CKCC were matched very well with a slight shift. The shift is caused by the $Ce^{3+}$ replaced with larger $La^{3+}$ ions.

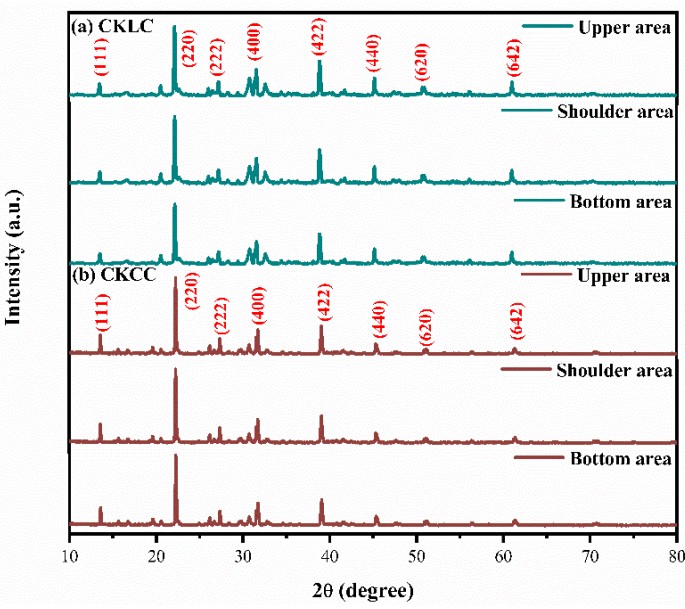

**Figure 2.** PXRD patterns of (**a**) CKLC and (**b**) CKCC crystals.

The TG-DTA curves of the CKLC and CKCC crystals are shown in Figure 3. The endothermic peak confirms that the melting points are 622 °C for CKLC and 549 °C for CKCC. In addition, no significant phase transition was observed until the melting point of the grown sample.

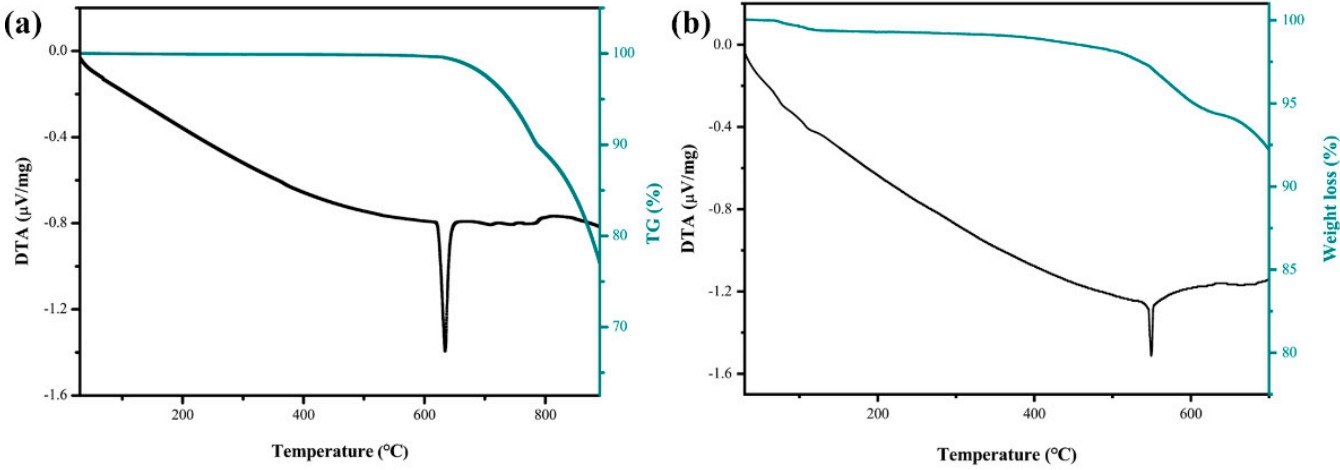

**Figure 3.** TG-DTA curves of (**a**) CKLC and (**b**) CKCC crystals.

### 3.3. Photoluminescence Spectra and Decay Time Curve

Photoluminescence excitation and emission spectra of CKLC (a) and CKCC (b) are shown in Figure 4. The ordinate represented intensity was normalized for better comparison. The excitation and emission bands of CKLC crystal could be fitted with five and two Gaussian compositions, respectively. For CKLC, five excitation peaks at 274, 281, 312, 339 and 356 nm corresponded to transitions from the $^2F_{5/2}$ level to three sublevels of $t_{2g}$ ($t_{2g}(1)$, (2) and (3)) and the two sublevels of $e_g$ ($e_g(1)$ and (2)) transitions of $Ce^{3+}$ ions, respectively. Two emission peaks at 370 and 403 nm corresponded to the transitions from the lowest sublevel of $t_{2g}$ to the $^2F_{7/2}$ and $^2F_{5/2}$ sublevels of $Ce^{3+}$ ions. In the same way, five excitation peaks were fitted with 2269, 278, 307, 343 and 352 nm in CKCC crystal, and two emission peaks were fitted with 372 and 401 nm in CKCC crystal. The stokes shifts of the $^2F_{5/2}$ to $t_{2g}$ (1) transition of $Ce^{3+}$ ions in CKLC and CKCC crystals were calculated to be 1063

and 1527 cm$^{-1}$, respectively. Note that both materials show some overlap of emission and excitation bands, so that Ce$^{3+}$ emission may be reabsorbed by another Ce$^{3+}$ center, which is similar to previous reports [19].

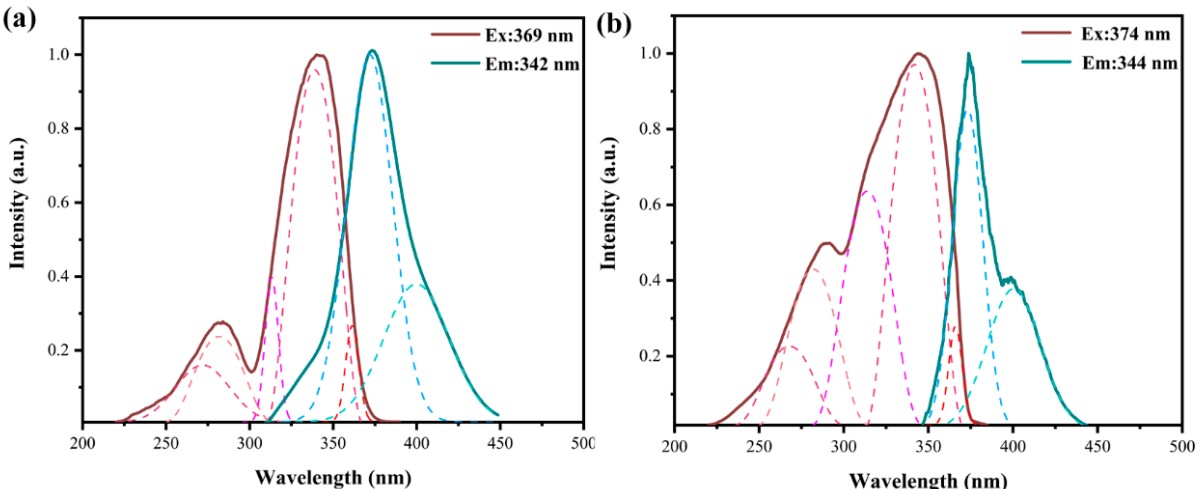

**Figure 4.** Normalized photoluminescence excitation and emission spectra of (**a**) CKLC and (**b**) CKCC crystals.

The photoluminescence decay curve of CKLC crystal using an excitation wavelength of 342 nm and a monitoring wavelength of 369 nm is displayed in Figure 5a. The curve could be well fitted by a single-exponential decay function. The decay time is 49.7 ns. Figure 5b shows the photoluminescence decay time curve of CKCC crystal. CKCC crystal has a shorter decay time than CKCC crystal. The decay time is 33.8 ns. Compared with the photoluminescence decay time of Cs$_2$LiLaCl$_6$:Ce crystal which is 36.6 ns, the photoluminescence decay time of CKCC is slightly shorter than that of Cs$_2$LiLaCl$_6$:Ce [16]. Compared with other crystals in the elpasolite crystal family, the performance of the photoluminescence decay time of the two crystals is satisfactory.

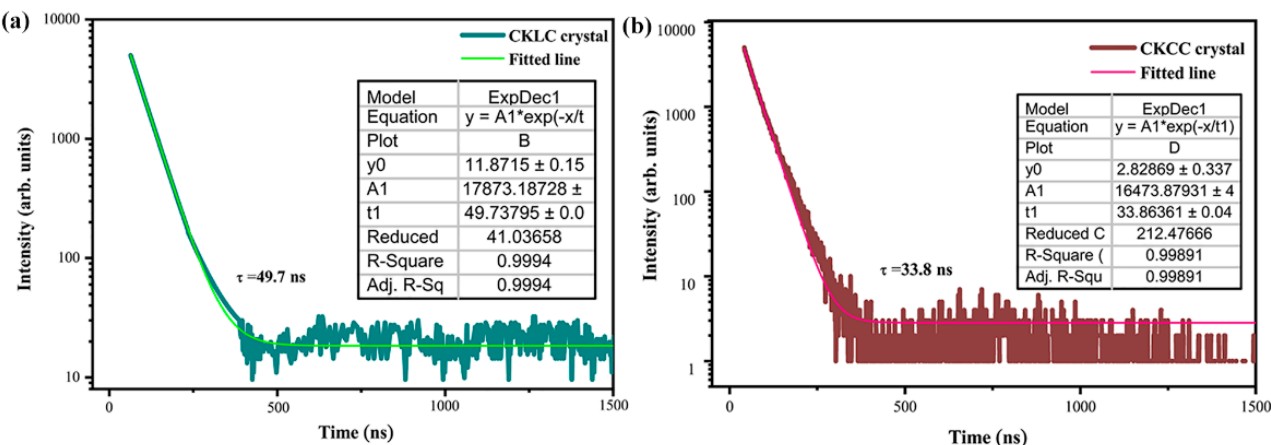

**Figure 5.** Photoluminescence decay time profiles of (**a**) CKLC and (**b**) CKCC crystals.

### 3.4. X-ray Induced Luminescence Spectra

Figure 6 shows the x-ray induced luminescence spectra of CKLC and CKCC crystals at room temperature. The typical Ce$^{3+}$ ions emission shows the double band shape due to 5d-4f transition, because the 4f ground state configuration yields two levels, via $^2F_{5/2}$ and $^2F_{7/2}$. However, the emission spectrum of both crystals shows a single band shape and contains almost a same broad band between 348 and 450 nm, peaking at 387 and 392 nm, respectively. This kind of peak often appears in crystals with high concentration of

$Ce^{3+}$ ions [20–22]. The red shift of the emission peaking in CKCC crystal was attributed to the increase in the concentration of doped $Ce^{3+}$ ions. In addition, the increase in the concentration of $Ce^{3+}$ ions also causes the intensity of the emission peak to become stronger.

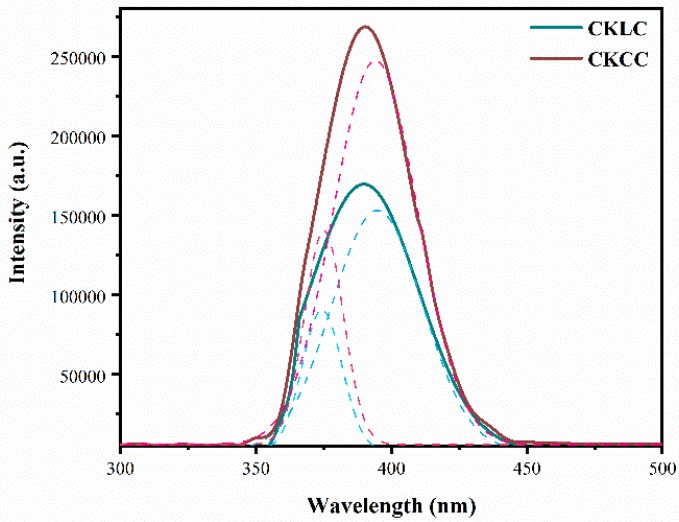

**Figure 6.** X-ray induced luminescence spectra of CKLC and CKCC crystals.

### 3.5. Pulse Height Spectrum and Scintillation Decay Curve

The pulse height spectra of CKLC and CKCC crystals were obtained under the excitation of $^{137}Cs$ source gamma rays, as shown in Figure 7. The energy resolution of the sample is calculated by Gaussian fitting to the full energy peak of the pulse height spectrum. The full energy peak under $^{137}Cs$ excitation of CKLC crystal can be observed in channel 656 and the energy resolution was 6.6%, which was worse than the reported 4.4% of $Cs_2NaLaCl_6$:4% $Ce^{3+}$ [17]. The CKCC crystal has an energy resolution of 5.2% at 662 keV and the full energy peak located in channel 546. It can be clearly concluded that the energy resolution of CKCC crystal is better than that of CKLC crystal.

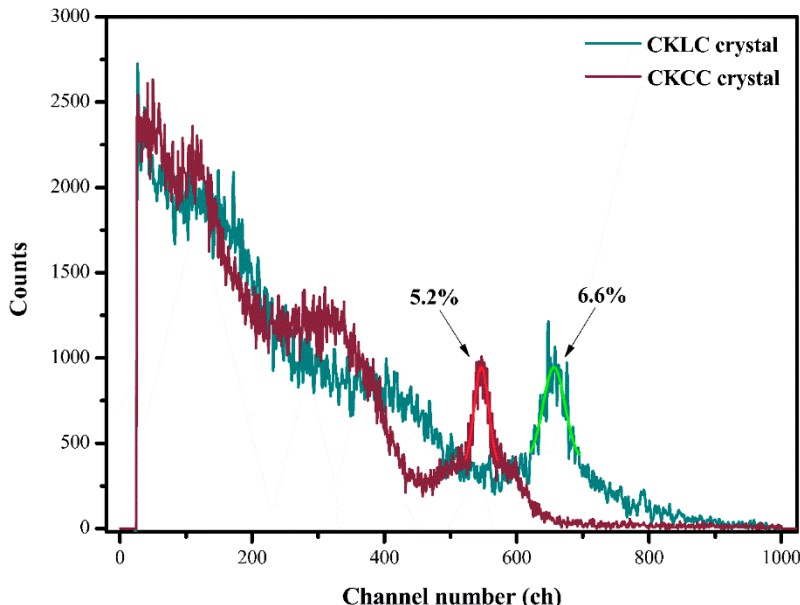

**Figure 7.** The scintillation pulse height spectrum of CKLC and CKCC irradiated with γ-rays from a $^{137}Cs$ source.

The scintillation decay curves of CKLC and CKCC crystal excited with γ-rays from a $^{137}Cs$ source are shown in Figure 8. To obtain the decay time constants, the average

decay curves were fitted with a double exponential function. In both crystals, two decay components are present. The scintillation light of $Ce^{3+}$ activated halide compounds is mainly composed of three parts: core valence luminescence (about 2 ns), prompt $Ce^{3+}$ ion luminescence (about 100 ns) and delayed $Ce^{3+}$ ion luminescence (about 1000 ns) [23]. The decay times of CKLC crystal (Figure 8a) are $\tau_1$ = 127 ns (33%) and $\tau_2$ = 1617 ns (67%), belonging to prompt and delayed $Ce^{3+}$ ion luminescence, respectively. The ultra-fast decay component was not observed, which may be due to the weakness of CVL [19].The decay times of CKCC crystal (Figure 8b) are $\tau_1$ = 2.86 ns (5%) and $\tau_2$ = 81 ns (95%), belonging to CVL and prompt $Ce^{3+}$ ion luminescence, respectively. The slower decay component was not observed. The CKLC and CKCC crystals are isomorphic crystals (elpasolite chlorides). Compared with CKLC crystal, the concentration of $Ce^{3+}$ in CKCC crystal reaches 100%, the distance between $Ce^{3+}$ becomes smaller in CKCC crystal. The shorter the distance between $Ce^{3+}$, the faster the electron hole pair migrates in the host lattice, which makes the delayed $Ce^{3+}$ ion luminescence smaller until it disappears. Compared with $Cs_2LiYCl_6$:Ce, $Cs_2LiLaCl_6$:Ce, $Cs_2NaLaCl_6$:Ce and $Cs_2NaLaBr_6$:Ce, CKCC has a faster scintillation decay time.

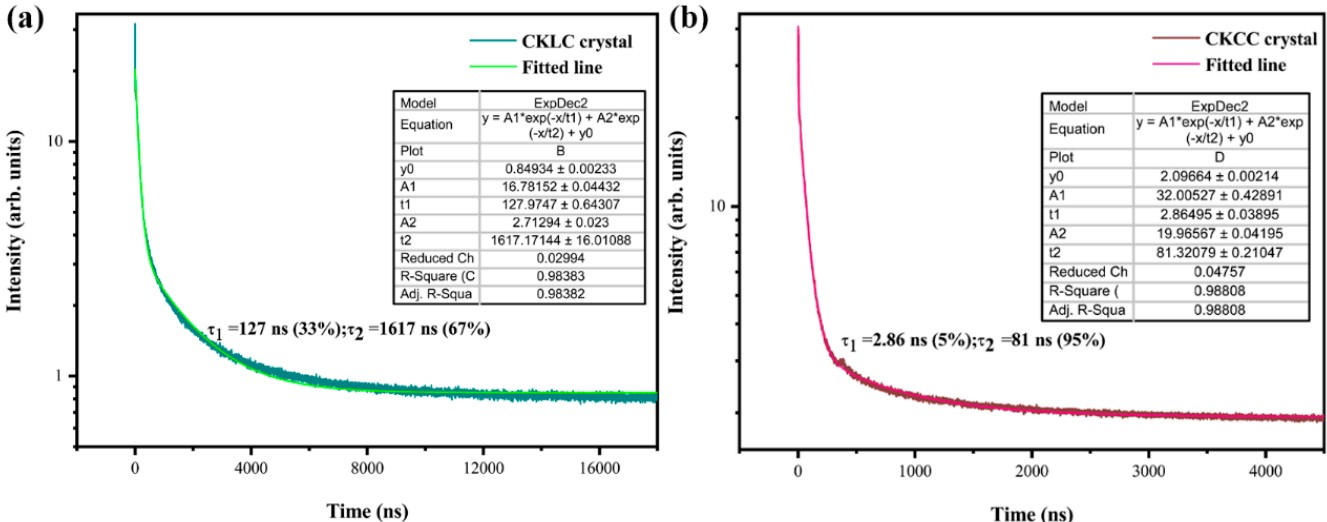

**Figure 8.** The scintillation decay time curve of (**a**) CKLC and (**b**) CKCC crystals measured under $^{137}$Cs $\gamma$-rays excitation at room temperature.

## 4. Conclusions

Single crystals of CKLC and CKCC were successfully grown from the melt by the vertical Bridgman method in sealed quartz ampoules. The luminescence and scintillator properties were measured at room temperature. The PXRD patterns showed both crystals have a cubic crystal structure with a higher effective Z-number. These indicated that they can easily obtain large single crystals and can effectively detect X-rays and $\gamma$-rays. The characteristics of luminescence and scintillator (such as energy resolution, decay time and emission intensity) indicate that CKCC crystal has smaller energy resolution, faster decay time and stronger emission intensity than CKLC crystal. Based on the above, this study shows that both materials are a promising scintillator in the field of radiation detection especially CKCC crystal.

**Author Contributions:** Conceptualization, H.W., S.P. and J.P.; methodology, H.W., S.P. and J.P.; investigation, H.W., J.X. and J.P.; resources, H.W., M.L. and J.P.; formal analysis, J.X. and J.G.; validation, H.W.; writing-original draft preparation, H.W.; writing-review and editing, H.W., S.P. and J.P.; supervision, S.P. and J.P.; Funding acquisition, S.P. and J.P. All authors have read and agreed to the published version of the manuscript.

**Funding:** This paper is partially sponsored by the National Natural Science Foundation of China (No. 511702190, 61775108 and 61875096). The authors are also thankful for the technical cooperation from Shanghai Institute of Ceramics, Chinese Academy of Sciences.

**Data Availability Statement:** The data presented in this study are available on reasonable request from the corresponding author.

**Conflicts of Interest:** The authors declare no conflict of interest.

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
