# Peer review of "Crystal Growth, Luminescence and Scintillation Characterizations of Cs2KLaCl6:Ce and Cs2KCeCl6"

_crystals, doi:10.3390/cryst11060653_

Round 1

Reviewer 1 Report

The manuscript under review presents results of characterisation of two new materials of elpasolite family from the viewpoints of their possible application as scintillation detectors. This is interesting topic that has always been high up on the agenda of material science but unfortunately the manuscript does not demonstrate the quality that is required for the publication in this journal. The quality of scientific results is average and there is nothing particularly exciting in the reported characteristics. I do not expect this paper to attract interest of many readers. Therefore, specialised scientific journal can be better choice for this study.

Below are a few comments that authors should consider before re-submission.

As a key material for the core components of high-end radiation detection àThis balloon statement overcrowded with the buzz words should be deflated. No key , no core , no high-end -  as they are meaningless words used merely for one purpose -artificially inflate the importance of the study. This is very primitive trick that should be exterminated.  

and have become a hot spot for 28 development à Bad expression as the crystals cannot be a hot spot .

The elpasolite crystal family has thousands of varieties due to the adoption of isomorphous replacement and solid solution methods, which makes the choice of scintillating materials 34 for various applications wider. àVery badly phrased

 In this report, we report on the crystal growth, luminescence and scintillation characterizations of newly CKLC and 37 CKCC single crystal. à very poor expression

At this point I stop commenting on the language blunders, but this should not be taken as endorsement of quality. It is rather opposite as the language is unforgivably poor and requires radical improvement.

Line 47 an ultra-dry quartz ampoule à provide quantitative definition of ultra-dry in comparison with dry.

Line 66 the software named MDI Jade 6.0 was devoted ???? to calculate the lattice 66 constant. Why the software is devoted to do this? Do you understand the meaning of this word?  

Line 73 A self-assembled X-ray generator with a tungsten 73 target was used to obtain the X-ray induced luminescence spectrum – A self-assembled X-ray generator  sounds very intriguing.  The X-ray-generators are commercial devises and if it is assembled by non-experts then I am very concerned about safety and then quality of this source and any data resulting from the measurements.

Line 80 Hama-matsu –Hamamatsu.

Line 105 The foggy opaque part of the crystal in the Fig. 1 is caused by being photographed 

in the crucible. – Is this photograph taken in the crucible or ampoule? What this passage is supposed to mean?  

Line 135 All discussion in this section should use nm (not cm-1) to be consistent with the data on X-ray luminescence.

Line 159 The decay curves should be displayed in Log scale.

Line  201 The same as above.

Reviewer 2 Report

Please find comments and suggestions in the attached file.

Reviewer 3 Report

The authors describe the synthesis and characterization of novel Ce3+-based halide scintillators for applications in the field of radiation detection. In particular, the authors focus their attention on the analysis of the structural and optical properties of these new solids, trying to highlight the differences that occur in the two samples. As a general comment, the manuscript is clear and the data are in principle solid and well discussed in the document. There are only a few comments that require revision by the authors, summarized here:

1) In the introduction section, it is not clear what is new in this study compared to what is described in the literature. The authors should clarify what they expect to achieve from these new materials compared to similar solids already studied in the past.

2) Regarding the CKLC sample, the Ce3+ load was set at 4%. Is there a specific reason stimulated by other findings or is it motivated by other articles?

3) While discussing the excitation and luminescence data, the authors comment on the presence of some bands using the wavenumbers (cm-1) for signal identification, but the x-axis in Fig. 4 is reported as the wavelength (nm). I think the authors should try to homogenize the language to make their considerations clearer. Furthermore, the existence of five excitation peaks highlighted by the authors in the text is not reflected in Fig. 4.

4) The PL lifetimes of CKLC and CKCC should be compared with other similar solids functionalized with Ce3+, in order to better appreciate the luminescence data.

Reviewer 4 Report

Review on article presented to Crystal: “Crystal growth, luminescence and scintillation characteriza-2 tions of Cs2KLaCl6:Ce and Cs2KCeCl6” by Haoyu Wang, Jianhui Xiong, Man Li, Jufeng Geng, Shanke Pan and Jianguo Pan*

The elpasolite crystal family has thousands of varieties. Current research focuses on discovering new scintillation 35 material belonging to the elpasolite crystal family. The single crystal of CKLC and CKCC was grown through the vertical Bridgman 52 technique. Detailed studies of obtained crystals quality were performed. the energy resolution was 6.6%, which was worse than the reported 4.4% of 177 Cs2NaLaCl6:4% Ce3+ [17].  So, a comparison with others crystals was made. I understood that the studied composition was not used before. Myself I did not find this composition in the internet. Presented conclusion that the obtained materials could be used as scintillators is acceptable.

I did not find a lot of errors. The only found are:

Ampouleby – line 102

Two  - line 145 uppercase  letter instead of lowercase letter

Therefore, the article could be published with minor corrections. I think that the article does not need next reviewing.

Round 2

Reviewer 1 Report

There is no visible improvement of language and presentation. Apparently, the reviewer  recommendation was not addressed at all.  Red highlighting of introduction is a futile attempt of misleading as the text is barely different from the original version. Unfortunately, in present form the manuscript does not match expectations of good paper.  Therefore, I again recommend sending this manuscript back for better revision involving expert language editor.  Bear in mind that I will repeat this recomendation untill revision is done properly.

Reviewer 2 Report

The manuscript are modified according to the comments and suggestions and are recommended for publication after english correction.